# MicroRNA-15a-5p mediates abdominal aortic aneurysm progression and serves as a potential diagnostic and prognostic circulating biomarker

Greg Winski [1,2], Ekaterina Chernogubova[1], Albert Busch[3], Suzanne M. Eken[1], Hong Jin[1,4],
Moritz Lindquist Liljeqvist[4], Tooba Khan[5,6], Alexandra Bäcklund[1], Valentina Paloschi[7,8], Joy Roy[4],
Rebecka Hultgren[4], Christine Brostjan[9], Gert J. de Borst[10], Joost P. G. Sluijter [11], Nadja Sachs [7,8],
Hans-Henning Eckstein[7], Reinier A. Boon [12,13,14], Joshua M. Spin [5,6], Philip S. Tsao[5,6],
Folkert W. Asselbergs[15,16,17] & Lars Maegdefessel[1,7,8] ✉

## Abstract

**Background** MicroRNAs are post transcriptional modulators of gene expression. We explored the diagnostic and prognostic value of circulating microRNAs in abdominal aortic aneurysm (AAA) disease, for which currently no established circulating biomarker is available.
**Methods** We profiled the expression of 754 human microRNAs in plasma from 187 patients with AAA and 190 matched non-diseased controls. To validate, we used two additional AAA patient cohorts, looking at circulating and aortic tissue-derived microRNA expression, and their correlation to AAA disease phenotype, as well as two murine AAA models.
**Results** We show that among 12 differentially expressed microRNAs, miR-15a and −659 are the most significantly up-regulated in AAA, whereas miR-1183 and -192 are the most significantly down-regulated. miR-15a is upregulated AAA patient tissues, and in plasma from two murine AAA models. In patients from three different cohorts, miR-15a expression levels in plasma, serum and aortic tunica media are significantly correlated with AAA diameter. Through modulation of miR-15a in human aortic smooth muscle cells, we identify several potential target genes of miR-15a known to be down-regulated in human AAA, suggesting its potential involvement in AAA pathology. Inhibition of miR-15a in vivo demonstrates a significant inhibition of murine aortic diameter growth at day 7.
**Conclusions** Our findings suggest that miR-15a is a potential biomarker of AAA. Through in vivo studies and based on its target profile, we show that miR-15a is involved in AAA pathogenesis and could help treatment, but also assist in risk-stratification of AAA patients and identify candidates for early AAA repair.

## Plain language summary

Abdominal aortic aneurysm (AAA) is a swelling in the aorta, which is the main blood vessel in the abdomen. If ruptured, this can cause severe bleeding. Biomarkers for abdominal aortic aneurysms (AAA) would allow more personalized and effective treatment. We compared the blood from people with and without AAA and identified a molecule, miR-15a, that was present at higher levels in people with AAA. Stopping the activity of miR-15a was found to reduce aorta swelling in a mouse model of AAA. Our results suggest miR-15a is involved in human AAA disease development and could potentially be used to indicate the presence and extent of AAA in humans.

Abdominal aortic aneurysm (AAA) is defined as a dilatation of the abdominal aorta, a diameter above 3 cm being considered aneurysmal. AAA affects 2–17% of men above the age of 65 with numbers varying depending on the sampled population[1,2]. While AAAs usually are clinically silent, AAA rupture has a mortality of around 75–90%[3], despite recent increases in success rate for ruptured AAA repair partly due to the introduction of endovascular aortic repair (EVAR)[4]. AAA size ≥55 mm has been correlated with rupture and is an indication for elective surgical repair[5], yet both open AAA repair (OR) and EVAR carry periprocedural (mainly OR) and/or long term (mainly EVAR) risk for complications[6]. Screening for AAA reduces

https://doi.org/10.1038/s43856-025-00892-w **Article**

ruptured-AAA (rAAA) and AAA-related mortality[7], but potentially leads to more unnecessary elective procedures and resultant complications[8], partly because rupture prediction based on imaging results alone is exceedingly difficult[9,10].

MicroRNAs (miRNAs) are a class of non-coding RNAs (ncRNAs) which fine-tune cellular responses[11]. Their deregulation is characteristic for a wide range of diseases, including cardiovascular disease and AAA. In recent years, we and other research groups have characterized several miRNAs (miR-21[12], miR-24[13], miR-29b[14–16], miR-195[17–20], miR-33a[21,22], miR-143/145[23] and miR-205/712[24]) as directly involved in AAA disease. MiRNAs, released by cells of the cardiovascular system, can be detected in the circulation and have been shown to be suitable as biomarkers[25]. To answer the question if miRNAs have the potential to outperform or complement classical cardiovascular disease biomarkers, sufficiently sized studies using multiple cohorts are required[25,26]. Of particular interest in our current era of personalized medicine approaches are biomarkers with prospective value, enabling clinicians to make decisions based on a potential exacerbation of disease as well as predicting the occurrence of future events. This is of great importance in selecting patients with already diagnosed small aortic aneurysms that might be at risk of rapid expansion and elevated risk of rupture[27]. In the present work, we used patient material from several longitudinal study cohorts and investigated different miRNAs with biomarker potential for AAA progression and rupture.

We have performed a first-of-its-kind sufficiently powered analysis of circulating miRNA biomarkers in plasma of AAA patients, looking at circulating miRNA signatures in a cohort of 400 individuals. To emphasize the translational prospect, we further studied potential targets not only associated with AAA disease, but also directly involved in the disease pathology. We identified miRNA-15a-5p (miR-15a) as a circulating biomarker of AAA and were able to show that its inhibition in experimental AAA models can slow disease progression. Furthermore, by correlating our findings with two additional AAA patient cohorts, we show that increased tissue and plasma/serum expression of miR-15a is strongly associated with a more severe AAA phenotype as indicated by greater aortic diameter. Based on these results we propose a plasma-based screening method that might contribute to a more accurate and biologically relevant assessment of AAA prevalence and risk, which is capable of better directing surgical or endovascular intervention to those patients who are most likely to benefit from timely AAA repair.

## Materials and methods
### Ethical approval
All collection of human materials was approved by the respective local ethic committees (Stockholm Regional Ethical Review board: 00-337, 2005/83-31, 2009/9-31/4, 2011/1863-32, 2012/916-31/4, 2018/176-32; Swedish Ethical Review Authority: 2019-04561, 2019-05050, 2019-05051; Ethikkommission der Technischen Universität München (Munich) 2799/10; Medical Research Ethics Committee (MREC) NedMec (Utrecht) 13-579; Ethikkommission der Medizinischen Universität Wien (Vienna) 1729/2014) and followed the guidelines of the Declaration of Helsinki. All human samples and data were collected after obtaining informed consent. Animal procedures conformed with EU and Swedish legislation concerning the protection of animals used for scientific purposes, and were performed under ethical approval of the local ethics committee (Swedish Board for Agriculture; Ethical permit no. N48/16).

### Power calculation
Based on calculations by the PASS software package (Department of Biostatistics, College of Public Health, UNMC, Omaha, NE), we calculated that a sample size of 200 cases (patients with an abdominal aortic diameter = AAD at baseline of >3 cm) and 150 controls (AAD < 3 cm) was needed to achieve 96% power for each miRNA, to detect a true difference in expression of at least 0.1 with estimated group standard deviations of 0.1 and 0.1 and with an experiment-wise error rate (EWER) of 0.0500 using a two-sided two-sample $t$ test.

### Study populations
**SMART (Second manifestations of ARTerial disease) cohort.** Rationale and design of the SMART study cohort, as well as abdominal aortic aneurysm (AAA) definition and policy, have been described in detail elsewhere[28,29]. Briefly, patients aged 44–80 with clinically manifest vascular disease or cardiovascular risk factors, referred to the University Medical Center Utrecht, The Netherlands, were enrolled. Patients received a standardized vascular screening including a health questionnaire, laboratory assessment, and ultrasonography. Based on the power calculation for this study, we included 200 SMART patients with ultrasonographically defined AAA (i.e., aortic diameter >40 mm), and 200 age-, gender-, CVD risk profile- and medication-matched non-AAA SMART control patients.

**Vienna AAA cohort.** The longitudinal observation of AAA patients was approved by the institutional ethics committee of the Medical University of Vienna (Ref 1729/2014) in 2014 and included serial blood drawings and CTA analyses at baseline and every 6 months for a maximum of 3 years. AAA patients without indication for surgical repair who presented at the outpatient clinic of the Division of Vascular Surgery, Vienna General Hospital were enrolled in the study. The exclusion criteria were recent (<1 year) tumor and/or chemotherapy, systemic autoimmune or hematological disease and organ transplantation. Written informed consent was obtained from each study participant and patient demographics were recorded by a structured questionnaire. From this study population, we gained access to serum samples from 28 patients, as well as their baseline and 6-month follow-up AAA diameter measurements.

**Stockholm AAA screening cohort.** The detailed study protocol has been described by Villard et al.[30]. In brief, subjects enrolled in the Swedish AAA-screening program, in which 65-year-old men are invited to undergo an ultrasound examination of the abdominal aorta, were recruited into this study between 2013 and 2019. Written informed consent was obtained from each study participant and patient demographics were recorded by a structured questionnaire. The plasma samples and the questionnaire were collected at the time of the screening ultrasound examination. AAA was defined as aortic diameter ≥30 mm by leading edge-to-leading edge measurement. Subjects not diagnosed with AAA (aortic diameter <30 mm) were sampled in a corresponding fashion and served as controls. Exclusion criteria were inadequate information provided in the structured questionnaire or unattainable blood samples. As part of this study, 68 plasma samples from AAA patients and 35 from non-AAA controls were analyzed.

**Stockholm AAA Biobank.** Patients undergoing open elective surgery for AAA at the Karolinska University Hospital were consecutively recruited into the Stockholm AAA Biobank. Sampling and processing were then performed as recently described by Lindquist Liljeqvist et al.[31]. From this study population, we gained access to already extracted RNA (miRNeasy Mini Kit, Qiagen, Hilden, Germany) stemming from 20 patients, as well as their plasma samples collected ahead of surgery. Separate samples were provided for the medial and adventitial tunicas.

**Limitations due to composition of study populations.** AAA is 4–6 times more common in men than in women, men have a 5–10-year earlier onset of disease and a continuously higher prevalence in all age groups. Population based screening is ongoing for 65-year-old men in Sweden. This enhances the possibility to include and analyze more patients with small aneurysms, however these will be men. Women are in minority both in the outpatient service but also in the treatment groups, since they are fewer and 5–10 years older. These impressive sex-differences do influence the biobanks collected worldwide in aneurysm cohorts. The included biobank material reflects these sex differences.

Nonetheless, this does limit the ability to generalize potential findings onto AAA pathology in women.

## Sample preparation

Blood samples were collected in non-treated (serum) and EDTA-treated (plasma) vials, centrifuged and subsequent supernatant stored at $-80\,°C$. Tissue was immersed in RNAlater (Ambion, Thermo Fisher Scientific, Waltham, MA, USA) immediately after sampling, kept in at $4\,°C$ overnight, and stored at $-80\,°C$ until RNA isolation. Tissues were lysed using a tissue homogenizer (ProScientific, Oxford, MS, USA). Extracted RNA was stored at $-80\,°C$ before analysis.

## In vivo AAA models

**Angiotensin II infusion**. As a model of AAA formation and rupture, we used angiotensin II (ANGII) infusion in 10 week-old male $Apoe^{-/-}$ mice (Taconic Biosciences, Hudson, NY, USA), first described by Daugherty et al.[32]. Briefly, osmotic minipumps releasing ANGII at 1000 ng/kg/min were placed under 2% isoflurane anesthesia. This ANGII dosage is reported to increase blood pressure by approximately 25 mmHg. In this model, AAA formation however does not seem to be caused by hypertension, since dampening as well as complete prevention of AAA formation has been achieved without blood pressure lowering[33,34]. At the pathological basis of AAA formation in this model is the mass influx of macrophages to the vascular medial layer and associated elastin degradation[35]. Using ultrasound (Vevo 2100, Visualsonics, Toronto, Canada), we measured the maximum infrarenal aortic diameter at baseline and at 7, 14, and 28 days after pump implantation. On day 7, blood was sampled in EDTA containers and centrifuged at $10,000 \times g$ for 10 min. On day 28, mice were sacrificed through $CO_2$ inhalation, exsanguinated by heart puncture, and perfused with $4\,°C$ PBS before organ harvesting. Aortas were embedded in OCT compound (Histolab, Gothenburg, Sweden), or snap-frozen, and stored at $-80\,°C$.

**Porcine pancreatic elastase infusion**. The porcine pancreatic elastase (PPE) infusion model, an established in vivo method for inducing infrarenal aortic aneurysms, was initially developed in rats by Anidjar et al.[36] and later adapted for use in mice by Pyo et al.[37]. In our study, we employed a modified version of the murine PPE model, visually detailed by Azuma et al.[38] and further optimized by Busch et al.[39].

In brief, 10-week-old male C57BL/6 mice (Charles River, Wilmington, MA, USA) were anesthetized with 2% isoflurane. A midline laparotomy was performed to access the abdominal cavity, and retractors were used to expose the retroperitoneum, with the intestines left in place. The aorta and inferior vena cava were then carefully separated from the region of the left renal vein to the aortic bifurcation. Aortic branches within 1 cm of the bifurcation were temporarily occluded using 6.0 silk sutures (Vömel, Kronberg, Germany).

Proximal and distal silk ties were applied, and an aortotomy was made using a 30-gauge needle tip. A heat-tapered PE-10 polyethylene catheter was inserted through the aortotomy and secured with a silk tie. The isolated aortic segment was infused for 8 min with PPE (1.5 U/mL in PBS; Worthington Biochemical Corporation, Lakewood, NJ, USA) under pressure sufficient to cause a 50–70% dilation of the vessel. Control animals received PBS only. Following catheter removal, the aorta was rinsed with saline and closed using a 10.0 polyamide monofilament suture (B Braun, Melsungen, Germany). All temporary ligatures were released, and the abdominal wall was sutured with 5.0 Vicryl (Ethicon, Somerville, NJ, USA), followed by skin closure. Post-operative analgesia was provided via 0.1 mg/kg buprenorphine.

Aortic diameter was measured via ultrasound imaging (Vevo 2100, VisualSonics, Toronto, Canada) pre-operatively and at 7, and 28 days post-infusion to assess aneurysm development. At the 4-week endpoint, mice were euthanized by $CO_2$ inhalation, exsanguinated via cardiac puncture, and perfused with cold ($4\,°C$) PBS. Aortic tissues were harvested,

embedded in OCT compound (Histolab, Gothenburg, Sweden) or snap-frozen, and stored at $-80\,°C$ for further analysis.

## In vivo miR-15a modulation

The method of miRNA modulation has been in detail described in previous publications[12,13,15]. Briefly, upon finishing of the Ang II osmotic pump implantation/surgical PPE instillation, while still under anesthesia, the mice received an intraperitoneal injection of 10 mg/kg miR-15a inhibitor, with control mice receiving a mismatch scrambled control (both Qiagen, Hilden, Germany).

**Ethical and statistical considerations**. In line with previous published work by our group and others, we used solely male mice for the purpose of this study. This limitation is due to limitations on the number of animals given the ethical constraints in our protocols. Assessing both male and female mice would have let to an increase in the total number of animals, which would have not been possible based on our available number of mice for this study. The required sample size was calculated based on previous experiments and modulation studies using these models. Studies had to be terminated once statistical significance was achieved. All animals were randomly assigned to a group by a blinded investigator. Age and body weights in between treatment and control were matched. Only animals from an identical genetical background were used for the study.

## Cell culture and transfection of human aortic smooth muscle cells

Primary human aortic smooth muscle cells (hAoSMCs, lot nr 3003) were purchased from Cell Applications (San Diego, USA) and cultured in complete Human SMC Growth Medium (Cell Applications; 311-500), supplemented with 1% PenStrep. Cells were seeded in 6-well plates at 125,000 cells/well and kept in growth medium overnight. Next morning, cells were starved for 24 h in OptiMEM (Gibco, ThermoFisher Scientific) + 2% FBS + 1% PenStrep. Shortly before transfection, medium was changed back to full growth medium as per above. Transfection of miRNA-modulators (miRVana miRNA mimic/inhibitor/scrambled control, ThermoFisher Scientific) was performed using Lipofectamine RNAiMAX (Thermo Fisher Scientific) in accordance with manufacturer's manual and using 7.2 µl RNAiMAX per well and a final modulator concentration of 50 nM. Cells were then cultured for 48 h. RNA was harvested as described in separate section, by pooling of two wells of each condition to assure adequate RNA concentration. Alternatively, cell kinetics were directly assessed using live-cell imaging. Briefly, starting 8–12 h after transfection, cells were continuously live imaged using the IncuCyte imaging system (IncuCyte Zoom, Sartorius AG, Goettingen, Germany) and their confluency automatically calculated using the built-in software.

## Tissue histology

**Sectioning**. Formalin-fixed paraffin-embedded (FFPE) blocks of aortic tissues from AAA patients/organ donor controls were received from the Munich Vascular Biobank. Sectioning was performed at 5 µm onto Superfrost Plus microscope slides. Paraffin was melted for 1 h at $60\,°C$. For FISH, melted sections were either processed directly or stored at $-80\,°C$, and all down-stream processing was done in RNAse-free conditions (autoclaved buffers/glassware). OCT-embedded mouse aortic tissues were sectioned at 8 µm onto Superfrost Plus microscope slides, dried at room temperature for 1 h and then stored at $-80\,°C$.

**Immunofluorescence (IF)/fluorescent in situ hybridization (FISH)**. To visualize miR-15a, double-DIG-labeled ISH probes from Exiqon (Qiagen, Hilden, Germany), specific either to miR-15a or a scrambled control, were used at 50 nM concentration. A protocol allowing for simultaneous FISH and IF was followed, described in detail by Nielsen et al.[40]. In brief, after FFPE sections were deparaffinized, or OCT-sections paraformaldehyde (PFA)-fixed (4% PFA for 10 min), peroxidase activity was quenched by incubation with 3% $H_2O_2$ for 10 min. Sections were then

predigested/permeabilized with proteinase-K in a concentration recommended by Exiqon for the respective section type. The ISH probes were hybridized for 2 h at 53 °C, after stringency washes were performed using serial dilutions of saline-sodium citrate (SSC) buffer (5x, 1x, 0.2x) at room temperature. Unspecific hybridization to other molecules was controlled for by using scrambled control probe in parallel. After blocking, sections were incubated with sheep-anti-DIG-POD (Sigma Aldrich) for 2 × 30 min, after which Alexa647-TSA substrate (Thermo Fisher) was applied for 10 min. Subsequently, a standard protocol for IF was followed. Briefly, after incubation with primary antibodies, secondary Alexa555-conjugated antibodies were used for signal visualization. Coverslips were mounted using Fluoromount-G Mounting Medium (Thermo Fisher). Imaging was performed using either a Leica Microsystems TCS SP8 (Wetzlar, Germany) confocal microscope or Olympus SLIDEVIEW VS200 (Tokyo, Japan) slide scanner.

### RNA isolation and RT-qPCR analysis

**Tissues/cells.** RNA extraction from tissues was performed using the Qiagen miRNeasy Mini or Micro Kits (Qiagen, Hilden, Germany), and from cells using miRNeasy Tissue/Cells Advanced Mini Kit (Qiagen), following the manufacturer's protocol. During extraction from cell lysates (intended for RNA sequencing) additional on-column DNAse digestion was performed. Total RNA concentration was assessed using a NanoDrop 2000 spectrophotometer (Thermo Fisher Scientific). Total RNA was diluted in RNAse-free water to a concentration of 2 ng/μL. Five μL (10 ng) was used for miRNA complementary DNA (cDNA) synthesis using the TaqMan MicroRNA Reverse Transcription Kit with reverse transcriptase primers for the selected miRNAs. For mRNA analysis, we used 500 ng of total RNA for cDNA synthesis using the TaqMan High-Capacity cDNA Transcription Kit with random primers. miRNA and mRNA expression were quantified by qPCR using TaqMan FAM- or VIC-labeled miRNA and mRNA assays. Data normalization was performed relative to U6 snRNA or *RPLP0*.

**Plasma/serum.** RNA extraction from plasma/serum was performed using Qiagen miRNeasy Serum/Plasma Advanced Kit (Qiagen) with *C. elegans* cel-miR-39 as spike-in control. To avoid technical bias, addition of spike-in was performed from one single stock dilution, and all samples within each study were processed as rapidly as possible, on the same day. A volume of 200 μl human plasma/serum (for mice 100 μl plasma diluted in 100 μl RNAse-free water) was mixed with 60 μl of Buffer RPL (supplemented with cel-miR-39) and the samples thereafter processed according to manufacturer's protocol. RNA was eluted in 20 μl RNAse-free water, of which 1 μl was then used for each downstream miRNA assay. TaqMan MicroRNA Reverse Transcription Kit (Applied Biosystems, Thermo Fisher Scientific) was used to synthesize cDNA. miRNA expression was quantified by qPCR using TaqMan FAM-labeled miRNA assays. Data normalization was performed relative to U6 snRNA or spike-in cel-miR-39 in human samples, and spike-in cel-miR-39 in mouse samples.

### High-throughput genomic analysis

**miRNA OpenArray.** miRNA was isolated from 200 μl of plasma using the TaqMan miRNA ABC purification kit (Thermo Fisher Scientific, Human Panels A and B). As a spike-in control, Ath-miR-159a oligonucleotide (synthesized by IDT; sequence: 5'-UUU-GGA-UUG-AAG-GGA-GCU-CUA-3') was added to each plasma sample. Purified miRNA was stored at −80 °C until further use. For reverse transcription, the Megaplex Primer Human Pools kit (Thermo Fisher Scientific) was employed, followed by preamplification with TaqMan PreAmp Master Mix (Thermo Fisher Scientific), in accordance with the manufacturer's standard protocols.

Following preamplification, samples were diluted using 0.1×TE buffer, formatted according to array layout, and loaded onto human microRNA panels using the OpenArray AccuFill system. TaqMan OpenArray real-time PCR was conducted using the QuantStudio 12 K Flex platform (Thermo

Fisher Scientific). Data were analyzed using R software. Quality control procedures included principal component analysis (prcomp in R) and assessment of data distribution (density in R).

A total of 758 miRNAs, including 4 endogenous controls, were profiled across all samples. Cycle threshold (Ct) values below 15 or above 35 were considered 'undetermined'. miRNAs not detected in over 50% of the samples were excluded from further analysis. To minimize technical variation and account for batch effects, quantile normalization was applied as described previously[41], eliminating the need for endogenous controls, which were subsequently removed from the dataset.

**RNA sequencing.** Cell culture, transfection and RNA extraction from lysates is described in separate sections. Extracted RNA from cell lysates stemming from hAoSMCs 48 h post-transfection with miR-15a-mimic ($n = 3$), miR-15a-inhibitor ($n = 3$) and scrambled control oligo ($n = 3$) was analyzed by a commercial vendor (Novogene UK Company Limited, Cambridge, United Kingdom) using following procedures. mRNA was purified from total RNA using poly-T oligo-attached magnetic beads. After fragmentation, the first strand cDNA was synthesized using random hexamer primers, followed by the second strand cDNA synthesis using dUTP. The directional library was complete after end repair, A-tailing, adapter ligation, size selection, amplification, and purification. The library was checked with Qubit and real-time PCR for quantification and Bioanalyzer for size distribution detection. The clustering of the index-coded samples was performed according to the manufacturer's instructions. After cluster generation, the library preparations were sequenced on an Illumina NovaSeq PE150 platform and >40 M paired-end reads were generated for each sample. Raw data (raw reads) of fastq format were cleaned by removing reads containing adapter sequence, reads containing poly-N and low-quality reads from raw data. At the same time, Q20, Q30 and GC content of the clean data was calculated. All downstream analyses were based on the clean, high-quality data. Paired-end clean reads were aligned to the reference genome using Hisat2 v2.0.5. FeatureCounts v1.5.0-p3 was used to quantify reads mapped to each gene. FPKM of each gene was calculated based on the length of the gene and its mapped reads. Differential expression analysis was performed using the DESeq2 R package (1.20.0). The resulting $p$ values were adjusted for false discovery rate using the Benjamini and Hochberg's approach. At this point, data was released by the commercial vendor and all subsequent analyses were performed independently by our group. Genes with an adjusted $p$ value ≤0.05 and absolute fold-change ≥1.5 were defined as differentially expressed.

### In silico prediction of AAA-relevant miR-15a targets

Through the National Center Biotechnology Information (NCBI) gene expression omnibus (GEO), we retrieved differential gene expression data from two previously published microarray studies on aortic tissue from AAA patients compared with controls (either autopsy[42] or organ donor[43]). In addition, we also retrieved microarray data from the recent study by Lindqvist Liljeqvist et al.[31] performed on AAA patient/organ donor tissues and divided into medial/adventitial tunicas either with or without intra-luminal thrombus (ILT). For miRNA target identification, we utilized in silico target prediction data from DIANA microT-CDS[44,45], miRDB 6.0[46,47], miRWalk 3.0[48], as well as experimentally validated targets from miRTarBase[49]. We performed an overlap analysis to select the databases that had good success in identifying genes deregulated in our miR-15a modulation experiments (Supplementary Fig. 1A, B). DIANA microT-CDS[44,45] and miRDB[46,47] were the in silico algorithms with the best performance and thus used for further analyses. As expected, miRTarBase, a collection of experimentally validated miRNA-gene interactions, performed similarly well, although its relatively smaller size compared to the above-mentioned in silico prediction tools meant a smaller absolute number of overlapping genes. Accuracy of target predictions was evaluated by comparing the significance of overlap between predicted/confirmed targets and genes down-regulated upon experimental miR-15a overexpression in vitro, using

**Table 1 | Patient characteristics human AAA cohorts**

| | Controls SMART ($n$ = 190) | AAA SMART ($n$ = 187) | AAA Vienna ($n$ = 28) | Controls Stockholm ($n$ = 35) | AAA Stockholm ($n$ = 68) |
|---|---|---|---|---|---|
| Aortic diameter | | | | | |
| Median (SD) | 16 (1.2) | 54 (10.0) | 49 (8.1) | 18 (1.9) | 40 (4.1) |
| Range | 13–17 | 42–110 | 31–68 | 15–23 | 35–49 |
| Age in years | | | | | |
| Median (SD) | 69 (7.9) | 69 (6.8) | 74 (8.6) | 65 (0.0) | 65 (0.0) |
| Range | 44–80 | 48–80 | 58–89 | 65–65 | 65–65 |
| Sex | | | | | |
| Male (%) | 178 (94) | 177 (95) | 25 (89) | 35 (100) | 68 (100) |
| Female (%) | 12 (6) | 10 (5) | 3 (11) | 0 (0) | 0 (0) |
| BMI | | | | | |
| Median (SD) | 26.2 (3.58) | 26.2 (3.41) | 27.0 (4.13) | 25.7 (3.16) | 28.1 (4.40) |
| Range | 18.8–42.6 | 19.0–38.8 | 19.3–37.6 | 21.1–34.6 | 19.2–41.7 |
| Smoking | | | | | |
| Current (%) | 34 (18) | 77 (41)**** | 9 (32) | 2 (6) | 23 (34)** |
| Ever (%) | 143 (75) | 176 (91)**** | 25 (89) | 16 (46) | 63 (93)**** |
| Risk factors | | | | | |
| Hypertension (%) | 108 (57) | 109 (58) | 25 (89) | 24 (69) | 40 (59) |
| Hyperlipidaemia (%) | 167 (89) | 174 (93) | 21 (75) | 5 (14) | 31 (46)** |
| CHD (%) | 84 (44) | 83 (44) | 9 (32) | 1 (3) | 19 (28)** |
| DM2 (%) | 50 (26) | 16 (9)**** | 8 (29) | 2 (6) | 20 (29)** |

*SD* standard deviation.

**$p < 0.01$ in Chi-square test vs. control group; ****$p < 0.0001$ in Chi-square test vs. control group.

hypergeometric test (also known as one-tailed Fisher's exact test). Significant overlap of genes present in target prediction databases with experimentally down-regulated genes was expected, whereas we expected no significant overlap with up-regulated genes which therefore served as internal controls of prediction accuracy.

### Statistics and reproducibility

Differences in RNA expression, measured by qPCR, were calculated as fold change versus control using the mean $\Delta Ct$ (defined as $Ct^{\text{target RNA}}$-$Ct^{\text{endogenous control}}$) within groups and compared using Student's $t$ test (unpaired, unless otherwise stated in figure legends). Two-tailed $p$ values were calculated, except for confirmatory/validation experiments with a predetermined hypothesis on the direction of change, where one-tailed $p$ values are appropriate. $p$ values < 0.05 were considered significant. Statistical analysis was performed based on measurements made in distinct samples. Replicates were defined as biological replicates. The miRNA OpenArray results were quantile normalized and analyzed for differential expression using R/Bioconductor libraries HTqPCR and limma. Multiple testing correction was performed using the Benjamini–Hochberg method. Linear regression was used to study the association between continuous variables and target RNA expression, and $t$-test was used to assess its significance. Pearson correlation coefficient was calculated to quantify the degree of correlation.

### Reporting summary

Further information on research design is available in the Nature Portfolio Reporting Summary linked to this article.

## Results

### Microarray analysis reveals 12 differentially expressed miRNAs in plasma of AAA patients

For our initial analysis, we used 203 human plasma samples from AAA patients versus 203 age-, medication-, and cardiovascular risk factor-matched controls from the Dutch Second Manifestations of Arterial disease

(SMART) cohort[28]. Patient and control characteristics are listed in Table 1. In total, expression of 754 miRNAs was assessed in 187 patients with AAA and 190 matched controls. Of all assayed miRNAs, 126 miRNAs were detected in >50% of the samples and were chosen for further statistical analysis. Principal component analysis revealed no clear clustering of samples (Supplementary Fig. 3). After correction for multiple comparisons, 12 miRNAs (Fig. 1A) were found to be significantly deregulated in AAA patient plasma. Among the most dysregulated miRNAs were the up-regulated miR-659 and miR-15a and down-regulated miR-1183 and miR-192 (Fig. 1B). In addition, miR-15a levels were found to be associated with larger aneurysms (>50 mm diameter, Fig. 1C and Supplementary Fig. 2A, B).

### miR-15a is deregulated in murine models of AAA

The vast majority of AAA miRNA biomarker studies to date have been largely correlative. We therefore chose to focus on miRNAs conserved between species, that could be further studied in experimental animal models of AAA. miR-15a and miR-192 were the top dysregulated miRNAs that are known to be conserved in rodents. In plasma of 10-week-old C57BL/6J *Apoe*$^{-/-}$ mice receiving angiotensin II infusion for 4 weeks, aneurysms were developed and miR-15a levels were confirmed to be significantly up-regulated, although no difference could be observed between the mice with ruptured vs. non-ruptured aneurysms (Fig. 1D). No significant differences in miR-192 levels could be detected in this model. To validate the effect on miR-15a levels, we used a second murine model, in which local AAA was induced with porcine pancreatic elastase (PPE) in 10-week-old C57BL/6J. In the PPE-model, miR-15a levels were most significantly up-regulated at day 7 post-aneurysm induction (Fig. 1E) and its expression was significantly correlated with relative diameter increase measured at this timepoint (Fig. 1F). This coincides with the time-point where the bulk of aortic diameter growth occurs (Fig. 1F), and after which the aortic tissue starts to recover after its initial disruption. At later time-points, expression of miR-15a declined and was significantly lower at day 56 compared to day 7, being at levels comparable to baseline (Fig. 1E). To study

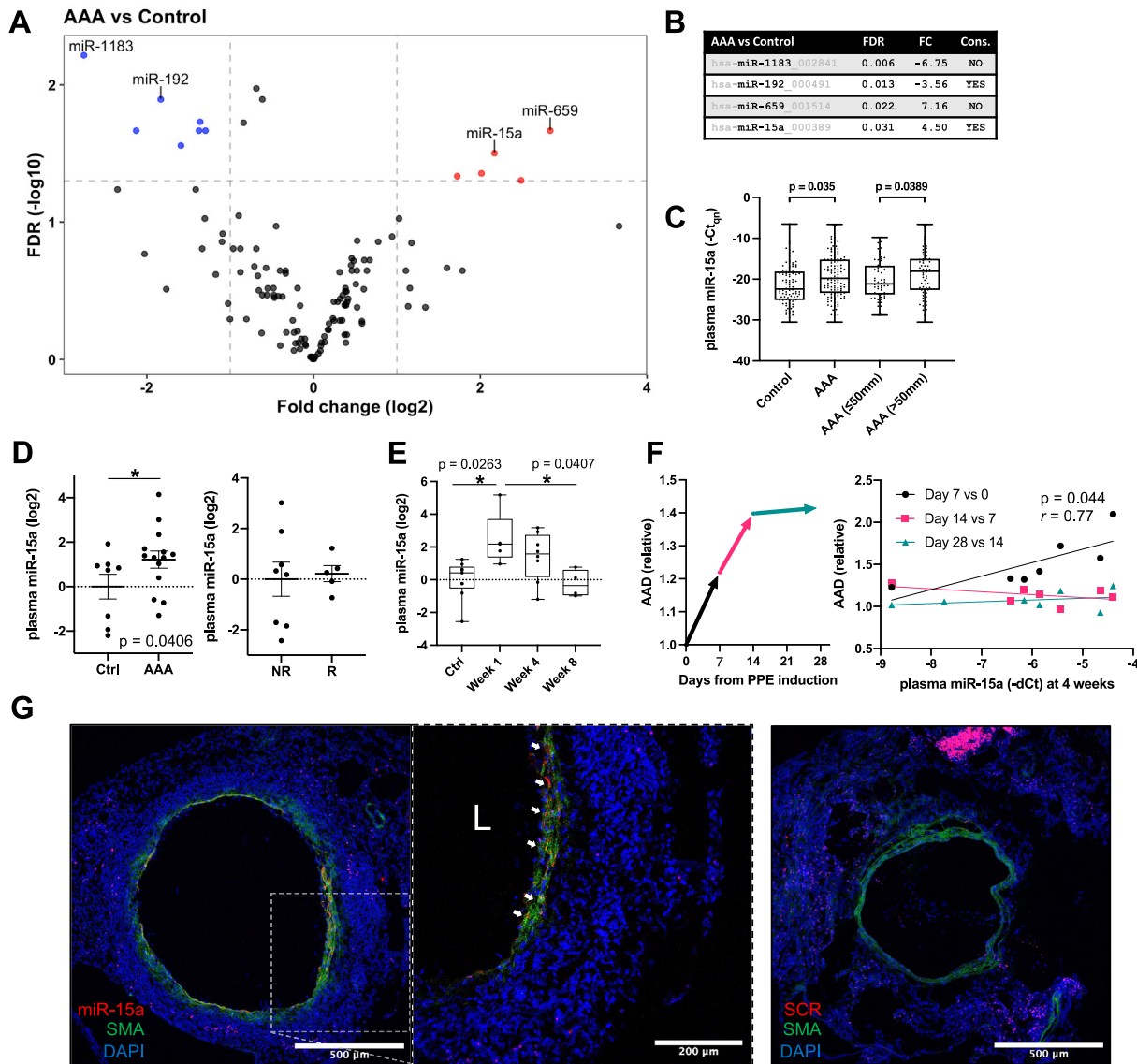

**Fig. 1 | miRNA expression in biofluids and aortic tissue of AAA model mice.**
**A** Volcano plot of differentially regulated miRNAs in patients ($n = 187$) compared to controls ($n = 190$) from the SMART cohort. Points above fold-change and FDR cut-off in gray are not conserved in mouse. **B** Table of top differentially regulated miRNAs in (**A**). Cons = conserved in mouse. **C** Quantile normalized Ct values ($Ct_{qn}$) of miR-15a in SMART control patients ($n = 92$), AAA patients ($n = 120$), AAA with diameter smaller or equal to 50 mm ($n = 63$), AAA with diameter bigger than 50 mm ($n = 57$). **D** Left: Expression of miR-15a in plasma of AngII model mice (AAA; $n = 13$) and saline controls (Ctrl; $n = 8$). Right: Expression of miR-15a in plasma of AngII model mice whose aortas were either ruptured/dissected (R; $n = 5$) or non-ruptured/non-dissected (NR; $n = 8$). Corresponding data for miR-192 can be found in Supplementary Fig. 1C. **E** Expression of miR-15a in plasma of PPE model mice sacrificed before PPE aneurysm induction (Ctrl; $n = 7$) or at different timepoints (Week 1, $n = 5$; Week 4, $n = 8$; Week 8, $n = 4$) post PPE aneurysm induction. **F** Left:

Progression of PPE model relative AAA diameter (AAd) as measured by ultrasound. Right: Correlation between relative AAd and plasma miR-15a upon sacrifice at 4 weeks. Colors correspond to time intervals as defined in the left panel. **G** Fluorescent in situ hybridization of miR-15a (red; Alexa647), immunofluorescence of α-SMA (SMA; green; Alexa555), fluorescently stained nuclei (DAPI) in mouse aortic tissue from the PPE model. The right-side box shows magnification of aortic wall with visible colocalization of medial α-SMA with miR-15a. Scale bars are 500 μm (left), 200 μm (middle), 500 μm (right). Limited unspecific hybridization can be seen in the adventitial layer, consistent with scrambled control probe hybridization shown in Supplementary Fig. 1D. L = artery lumen. Data points are presented as Mean ± SEM. Differences between means were analyzed using unpaired Student's *t* test. *p* values in **A** were controlled for false discovery rate (FDR) using Benjamini–Hochberg correction. For analysis of correlation Pearson correlation coefficient (*r*) was calculated. AAA aortic abdominal aneurysm.

the localization of miR-15a expression, we performed fluorescent in situ hybridization in PPE model mouse aortas (harvested at day 14). Expression of miR-15a co-localized with α-SMA, a smooth muscle cell specific marker (Fig. 1G).

## miR-15a is expressed in SMCs of the medial layer and ECs of adventitial vasa vasorum

Given the findings in PPE model mice, we performed fluorescent in situ hybridization of miR-15a in tissues of AAA patients and organ donor controls. Expression of miR-15a again co-localized with α-SMA, a smooth

muscle cell marker (Fig. 2A). A clear signal could be observed within the medial layer of aortas, as well as in the endothelial layers of the micro-vessels supplying the adventitial layer (*vasa vasorum*). Different roles of miR-15a in endothelial cells have been previously described[50,51]. In a AAA context, Gao et al. have reported a pro-apoptotic role of miR-15a-5p on VSMCs— through inhibition of CDKN2B, which was consequently found to be down-regulated in AAA patient tissues[52]. In fact, several miRNAs have previously been shown to regulate AAA through their effects on SMC dynamics[12,13,23,53,54]. As expected, upon transfection of hAoSMCs with miR-15a mimics, we observed clear attenuation of cell growth compared to

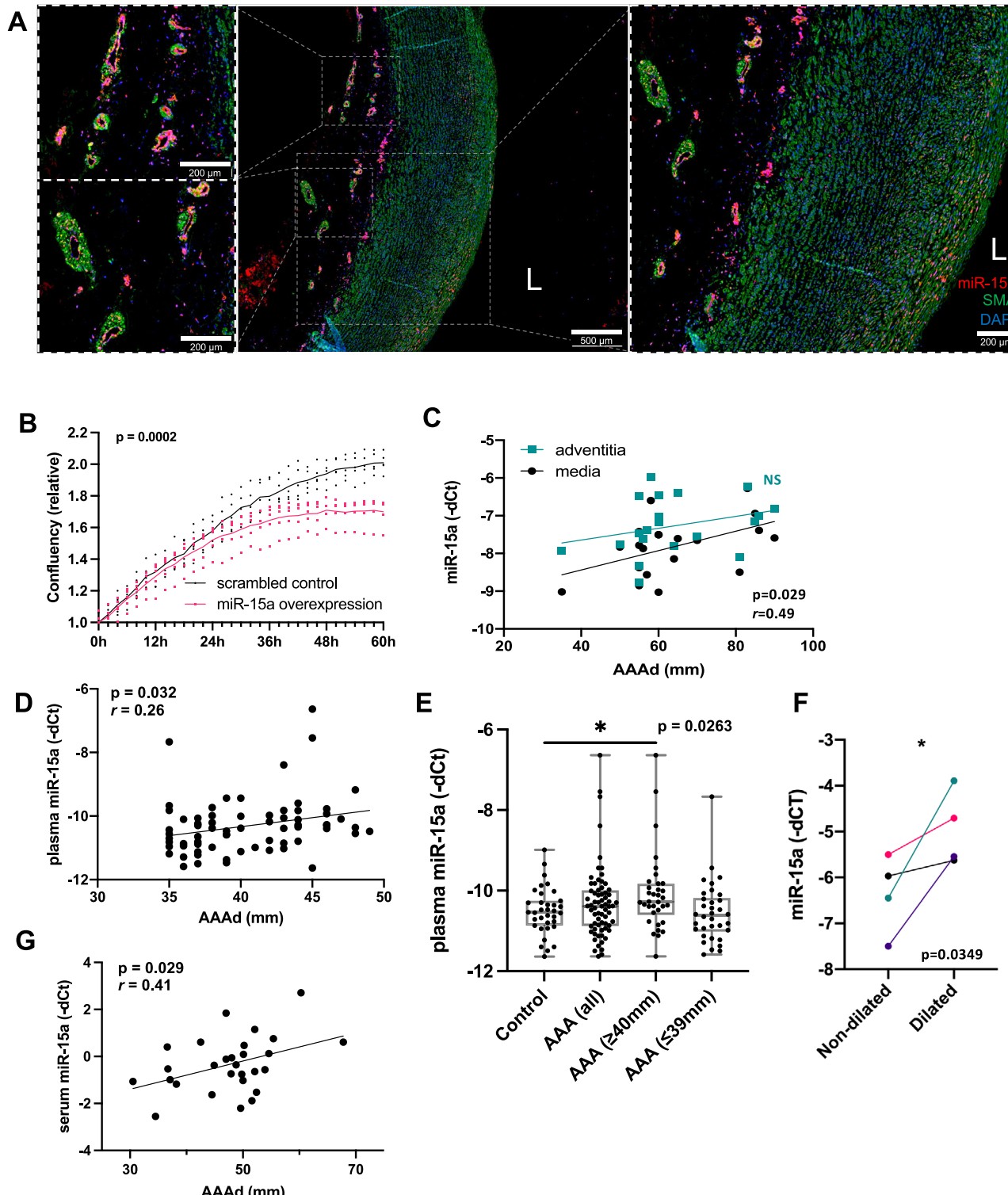

scrambled mimic transfected controls (Fig. 2B), with opposite effect using miR-15a inhibitors previously having been shown by Gao et al.[52].

## Levels of miR-15a in circulation and in aortic tunica media are correlated with AAA diameter

Next, we sought to verify the up-regulation of miR-15a levels and its involvement in AAA by qPCR-based measurements in independent AAA patient cohorts. For this purpose, we utilized samples from two cohorts of AAA patients based in Stockholm (Stockholm AAA Biobank). The first

cohort consisted of tissue samples (medial/adventitial layers) of patients undergoing open AAA repair, as well as their respective plasma samples collected ahead of surgery. We found medial, but not adventitial, expression of miR-15a to be significantly associated with AAA diameter (Fig. 2C), which given its described pro-apoptotic effects on VSMCs[52] pointed towards a direct involvement in AAA pathology.

The second cohort consisted of patients with AAAs discovered as part of the Swedish AAA screening program (65-year-old males). We found that their plasma miR-15a levels significantly positively correlated to the AAA

**Fig. 2 | miR-15a expression in human aorta and biofluids from validation cohorts and its effects on hAoSMC growth in vitro. A** Fluorescent in situ hybridization of miR-15a (red; Alexa647), immunofluorescence of α-SMA (SMA; green; Alexa555), fluorescently stained nuclei (DAPI) in human aortic tissue. The right-side box shows magnification of aortic medial tunica with visible colocalization of α-SMA, left-side boxes show magnifications show high levels of miR-15a expression within the aortic tunica intima (on the lumen side of α-SMA signal). L = artery lumen. Scale bars are 200 µm (left; top and bottom), 500 µm (middle), 200 µm (right). **B** Live-time cell imaging analysis of relative cell confluency in hAoSMCs treated with either miR-15a mimic (overexpression, $n = 5$) or control mimic (scrambled control, $n = 5$). **C** Correlation of tissue miR-15a expression in tunica adventitia (adventitia) and tunica media (media) of AAA patient aortas (Stockholm open AAA repair patients;

$n = 20$) and their AAA diameter (AAAd). **D** Correlation of plasma miR-15a expression in AAA patients (Stockholm AAA screening cohort; $n = 68$) and their AAAd. **E** Expression of miR-15a in plasma of AAA patients (Stockholm AAA screening cohort; $n = 68$ for AAA of which $n = 34$ for AAA $\geq 40$ mm and $n = 34$ for AAA $\leq 39$ mm) and matched controls ($n = 35$). **F** Expression of miR-15a in aortic tissue from AAA patients (Munich Vascular Biobank; $n = 4$) taken either from dilated or non-dilated part of the aorta. **G** Correlation of serum miR-15a expression in AAA patients (Vienna AAA cohort; $n = 28$) and their AAAd. Data points are presented as Mean ± SEM. Differences between means were analyzed using unpaired Student's $t$ test (except **G**, where paired Student's $t$ test was used). For analysis of correlation Pearson correlation coefficient ($r$) was calculated. AAA aortic abdominal aneurysm, L lumen.

diameter (Fig. 2D). We observed no significant difference in miR-15a expression between AAA patients and the control group (Fig. 2E), possibly because these patients had smaller aneurysms compared to patients in the discovery (SMART) cohort (Table 1) and suggesting that the miR-15a expression in SMART patients was connected to AAA diameter (Fig. 1C). When solely looking at patients from the Swedish AAA screening program cohort with AAA diameter ≥40 mm (half of the cohort), miR-15a levels in plasma were indeed significantly higher than in control patients (Fig. 2E and Supplementary Fig. 2C). Next, we sought to verify the findings in a separate cohort of AAA patients from Vienna, Austria. In this longitudinal follow-up cohort of known AAA patients, they had relatively larger AAA diameters compared with the newly diagnosed AAA patients from the above-mentioned Swedish AAA screening program. In serum samples of these patients, we observed that the levels of miR-15a significantly correlated with their respective AAA diameters (Fig. 2G), but not with diameter increase during the 6-month follow-up.

Given the fact that levels of both medial and circulating miR-15a expression could be connected to AAA phenotype, we wanted to understand whether their circulating presence in fact originated from the aneurysmal site, or if the increased in situ expression of tissue miR-15a was more reflective of its overall transcriptional activity across different organs. For this purpose, we analyzed tissue samples of patients undergoing open AAA repair. We hypothesized that if the increase of miR-15a levels in circulation was to be caused by AAA disease progression, spatial expression differences between aneurysmal and non-aneurysmal regions of the aorta were to be expected. From patients undergoing open AAA repair, we were able to collect aneurysmal tissue, as well as aortic tissue from the proximal region of healthy aorta. Paired analysis of these samples revealed that the diseased tissue regions of these patients' aortas had significantly higher miR-15a expression (Fig. 2F), strengthening the hypothesis that circulating miR-15a could be originating from these regions.

### Inhibition of miR-15a attenuates murine AAA growth

To better understand whether miR-15a is involved in AAA pathogenesis or merely indicative of underlying pathologic processes we performed intervention experiments in established PPE and ANGII murine models. PPE-model mice receiving a miR-15a inhibitor displayed less aortic diameter growth at all measurement timepoints post-induction (Fig. 3A). No significant effect could be observed in ANGII-model mice (Fig. 3B), however we observed that in contrast to the control animals, the aortas of miR-15a inhibitor treated mice did not increase in size enough to be considered aneurysmal (defined as 1.5-fold diameter increase). Aortic sections of PPE mice which received the miR-15a-inhibitor had distinctly higher smooth muscle actin content and more preserved elastic layers compared to scrambled miR-treated mice (Fig. 3C).

### Overexpression of miR-15a in hAoSMCs leads to broad gene expression changes associated with decreased SMC viability and AAA disease

Based on our findings thus far, we hypothesized that miR-15a could be directly involved in AAA pathology through detrimental effects on VSMCs.

As noted above, several miRNAs have previously been shown to regulate AAA through their effects on VSMC[12,13].

Among in silico predicted or experimentally confirmed miR-15a targets are many genes of relevance to AAA pathology and known to be down-regulated in AAA conditions. To better understand which of these targets could be linked to the observed detrimental effects of miR-15a on SMC dynamics in vitro, we performed RNA-sequencing of hAoSMCs transfected with miR-15a-modulators (mimic, inhibitor, and scrambled control). Upon transfection with miR-15a-mimic, 1030 genes were significantly down-regulated and 741 were significantly up-regulated (Fig. 4A), whereas miR-15a-inhibition led to 23 down-regulated and 82 up-regulated genes (Fig. 4B).

These changes are likely to combine direct effects by miR-15a, however many of them likely through downstream changes in gene expression rather than their direct modulation by miR-15a per se. To make the approach more stringent, we took advantage of miRNA target prediction databases and combined different resources to define their overlapping predictive targets (Supplementary Fig. 1A, B).

Furthermore, we used AAA gene-expression datasets by Lenk et al.[42], Biros et al.[43] (both of human AAA tissues) and Lindquist Liljeqvist et al.[31] (human AAA tunica media with or without intraluminal thrombus) to identify potential down regulated gene targets relevant to human AAA disease. Looking selectively at genes previously described as down-regulated in AAA, 204 were down-regulated and 104 were up-regulated upon miR-15a-mimic transfection (Fig. 4C), whereas 5 were down-regulated and 11 were up-regulated upon miR-15a-inhibition (Fig. 4D).

To better understand the overall impact of these changes, we performed a gene-set enrichment analysis (GSEA). Using MSigDB-H (hallmark) gene-sets, we discovered that transfection with miR-15a-mimic led to overall increase of genes involved in inflammatory response, apoptosis, hypoxia, and mesenchymal transition and decrease of cell-cycle related genes (Fig. 5A). In addition, a DisGeNET-centered analysis identified an increase of genes associated with AAA and other vascular diseases (Fig. 5E).

Due to the nature of miRNA function, genes simultaneously down-regulated upon miR-15a overexpression and up-regulated upon its inhibition had the highest overlap with experimentally validated or predicted targets of miR-15a (Fig. 5B, C). In total, 183 genes were regulated in this pattern (Fig. 5B), of which 60 were significantly down-regulated in AAA datasets (Fig. 5C). Of these, 19 are in silico predicted and 12 are experimentally confirmed targets of miR-15a (Fig. 5D). Some of these genes are discussed in the sections below.

In summary, we were able to show a VSMC-relevant target profile of miR-15a of importance to human AAA disease. As it is a typical feature of miRNAs to regulate functional gene-networks rather than specific genes[55], we believe that the role of miR-15a is more complex than inhibition of a specific key AAA gene. For the purpose of biomarker discovery, the broad involvement of miR-15a in AAA-relevant processes could yield important insight into which patients are at-risk for disease development and/or rapid progression.

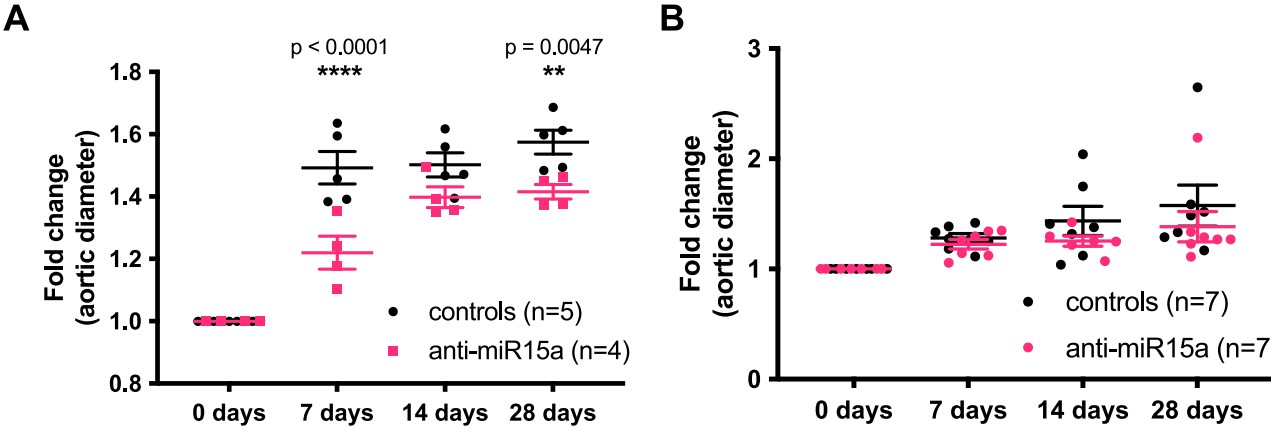

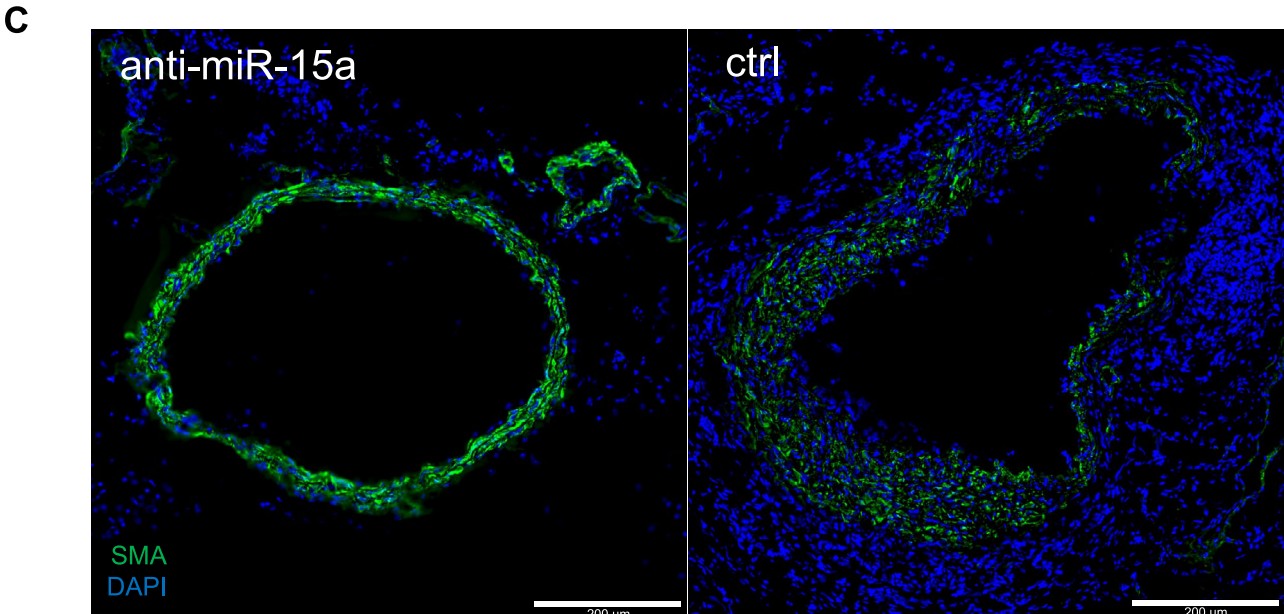

**Fig. 3 | Effects of miR-15a modulation in murine AAA. A** Time-course of relative change in aortic diameter in PPE model mice treated with miR-15a inhibitor (anti-miR-15a; $n = 4$) or PPE model mice treated scrambled control inhibitor (control; $n = 5$). **B** Time-course of relative change in aortic diameter in AngII model mice treated with miR-15a inhibitor ($n = 7$) or scrambled control inhibitor (control; $n = 7$). Data are presented as Mean ± SEM. Differences between means were analyzed using multiple Student's *t* tests and controlled for multiple comparison using Benjamini–Hochberg's method. **C** Immunofluorescence of α-SMA (SMA; green; Alexa647), fluorescently stained nuclei (DAPI) in mouse aortic tissue from the PPE model treated with either miR-15a inhibitor (anti-miR-15a) or scrambled control inhibitor (ctrl). Scale bars are 200 μm. Data points are presented as Mean ± SEM.

## Discussion

Screening for AAA, with its silent development, progression, and significant risk for an acute, lethal outcome, has long been a topic of discussion. The combined evidence suggests that the patient group most at risk, >65-year-old males, do benefit from screening[7,56]. This benefit is even more pronounced among smokers[57]. According to a recent WHO report, globally, tobacco use has decreased by roughly a third during the past 20 years[58]. Given that smoking is the major risk factor for AAA disease prevalence and progression[59,60], one would expect a decreasing benefit-to-risk ratio of population-wide screening approaches in countries/regions with decreasing cigarette use. Screening for AAA, usually performed by ultrasound, is non-invasive, quick, and relatively inexpensive, leading to its broadly reported cost effectiveness in preventing AAA related mortality[61]. As AAA remain largely asymptomatic, the major reason for their repair is rupture prevention. However, even with ultrasound screening, prediction of AAA rupture remains a challenge in the clinical setting. As surgery currently remains the only option of preventing rupture, it is the peri- and post-procedural risk, rather than the means of screening procedure itself, that contributes to the

risk aspect of the benefit-to-risk calculations. Surgery should thus only be provided when a rupture risk above a certain threshold. The large body of evidence on patient selection for AAA surgery is, due to ethical constraints, based on correlative retrospective studies[62]. Currently, the best tool for risk stratification is continued measurement of AAA diameter, where a diameter >55 mm (in men) and/or rapid growth (>10 mm/year) can qualify for surgery[62]. A better understanding of which molecular processes drive AAA pathology can not only help to develop the first non-surgical treatment alternatives, but perhaps more importantly lead to novel methods of risk-stratification. Here, discovery of novel biomarkers would play an important role, especially if these could be shown to be directly involved in the pathological processes surrounding the disease. We show that to achieve better allocation of AAA repair surgery, and to prevent death due to AAA rupture, miRNAs represent a promising group of biomarkers able to give a signature of AAA development on a cellular level (Fig. 6).

Previous studies searching for AAA biomarkers have shown promise, albeit often hindered by small sample sizes and heterogenic threshold values[63]. When it comes to studies on miRNA biomarkers, major challenges

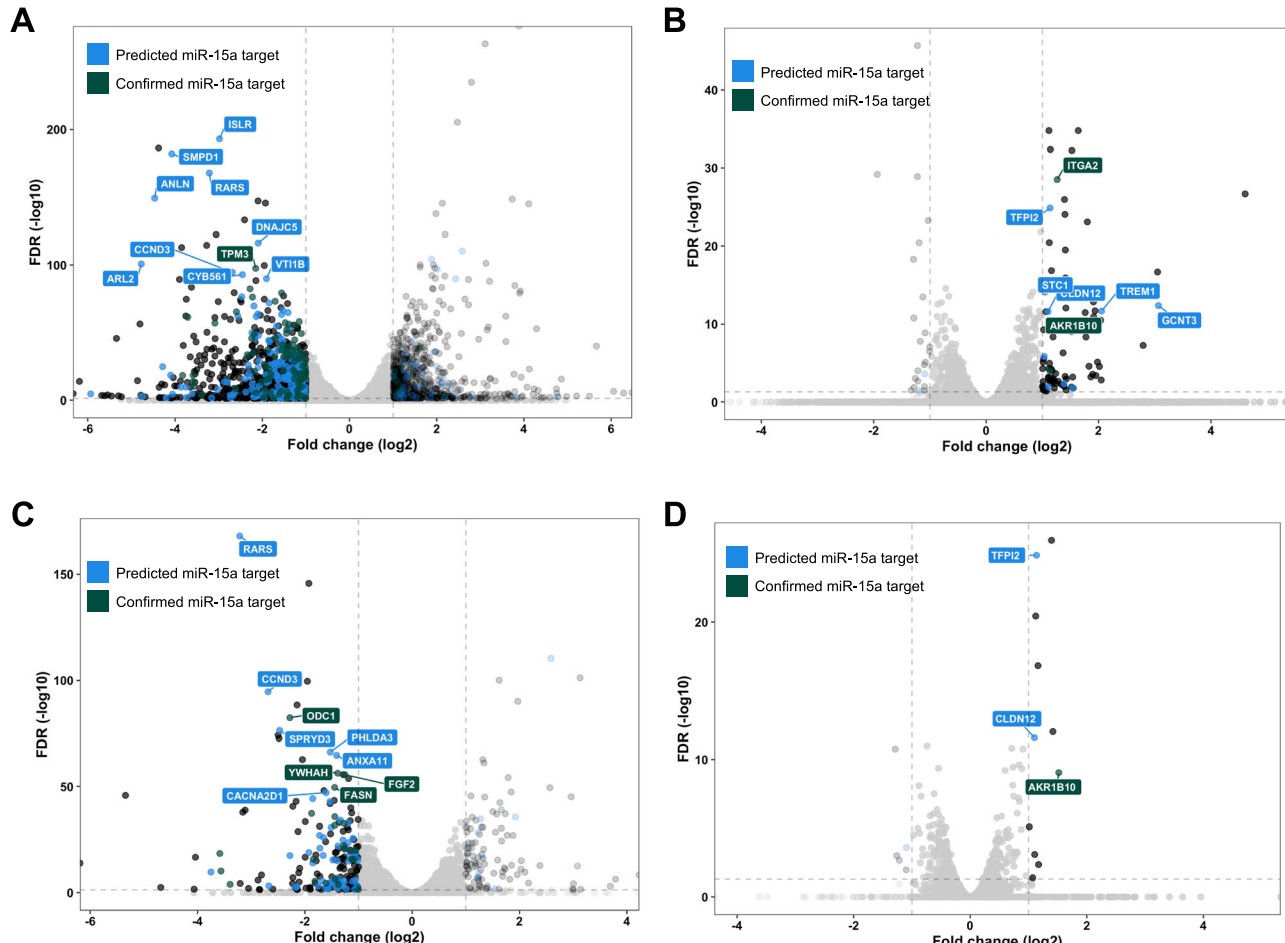

**Fig. 4 | Effects of miR-15a modulation in hAoSMCs on gene expression.**
**A** Volcano plot of significantly down-regulated (FDR < 0.05, fold change (log2) < −1) genes in hAoSMCs transfected with miR-15a mimic. Highlighted in blue are predicted miR-15a targets (present in DIANA or miRDB), in green experimentally confirmed miR-15a targets (present in miRTarBase). Labeled are top 10 most-significantly down-regulated genes that are confirmed/predicted targets of miR-15a. **B** Volcano plot of significantly up-regulated (FDR < 0.05, fold change (log2) > 1) genes in hAoSMCs transfected with miR-15a inhibitor. Genes are highlighted as in (**A**). Labeled are top 7 most-significantly up-regulated genes that are confirmed/predicted targets of miR-15a.

**C** Volcano plots as shown in (**A**). Genes not down-regulated in human AAA datasets (mean fold change < −1.5 in datasets by Biros, Lenk, tunica media by Lindquist Liljeqvist) have been removed. Genes are highlighted as in (**A**). Labeled are top 10 most-significantly down-regulated genes that are confirmed/predicted targets of miR-15a. **D** Volcano plots as shown in (**B**). Genes not down-regulated in human AAA datasets have been removed, all genes are can be found in Supplementary Fig. 1B. Genes are highlighted as in (**A**). Labeled are top 3 most-significantly up-regulated genes that are confirmed/predicted targets of miR-15a.

in interpretation of results have stemmed from factors such as insufficient sample sizes and confounding patient characteristics[64], as well as methodological issues inherent to miRNA measurement such as different normalization strategies[65]. The present work attempts to overcome these issues by including a power analysis, use of standardized sampling methods, and providing internal validation through the inclusion of different patient cohorts. In addition, we have tried to focus and shed light on miRNAs that, apart from being dysregulated in circulation, also seem connected to the underlying pathological processes, with the belief that these miRNAs can not only help in diagnosis of AAA, but also in risk stratification of AAA patients.

We are not the first ones to report on the role of miR-15a in aortic pathology. Dong et al.[66] found miR-15a to be up-regulated in plasma of patients with acute aortic dissection. In the context of AAA, several published studies have described it as down-regulated in AAA whole blood/plasma—in contrast to our reported findings. Kin et al.[67] have reported miR-15a as up-regulated in tissues of AAA patients, and down-regulated in AAA patient plasma compared to healthy controls, although not when compared to cardiovascular disease controls. In that study, miR-16 was used as normalization control. This was likely not ideal since miR-15a stems from the

same miRNA-family as miR-16 and both are transcribed from the same cluster on chr13. Stather et al.[68] reported miR-15a as down-regulated in whole blood of AAA patients when compared with healthy controls, but not when compared to peripheral artery disease (PAD) controls. Other authors have subsequently misquoted the reports of Spear et al.[69] on down-regulation of miR-15a-3p in plasma of AAA patients compared to PAD controls, as opposed to miR-15a (miR-15a-5p). Finally, Wanhainen et al.[70] found miR-15a-5p to be slightly down-regulated in their qPCR-based study of plasma from a big cohort of AAA patients and controls. They employed a spike-in based normalization protocol, which may be sensitive to batch effects and other types of bias[71]. In fact, we found no significant correlation between the full dataset published in the aforementioned study and data from our discovery cohort, further underlining the possible impact of differences in methodology.

Overall, we believe this strengthens the argument that certain design limitations can lead to difficulties when interpreting study results. To counter this, our study employed a large and balanced discovery cohort together with a robust quantile normalization method. We also present data from two independent validation cohorts, consisting of both small and larger aneurysms, showing association between increased circulating miR-

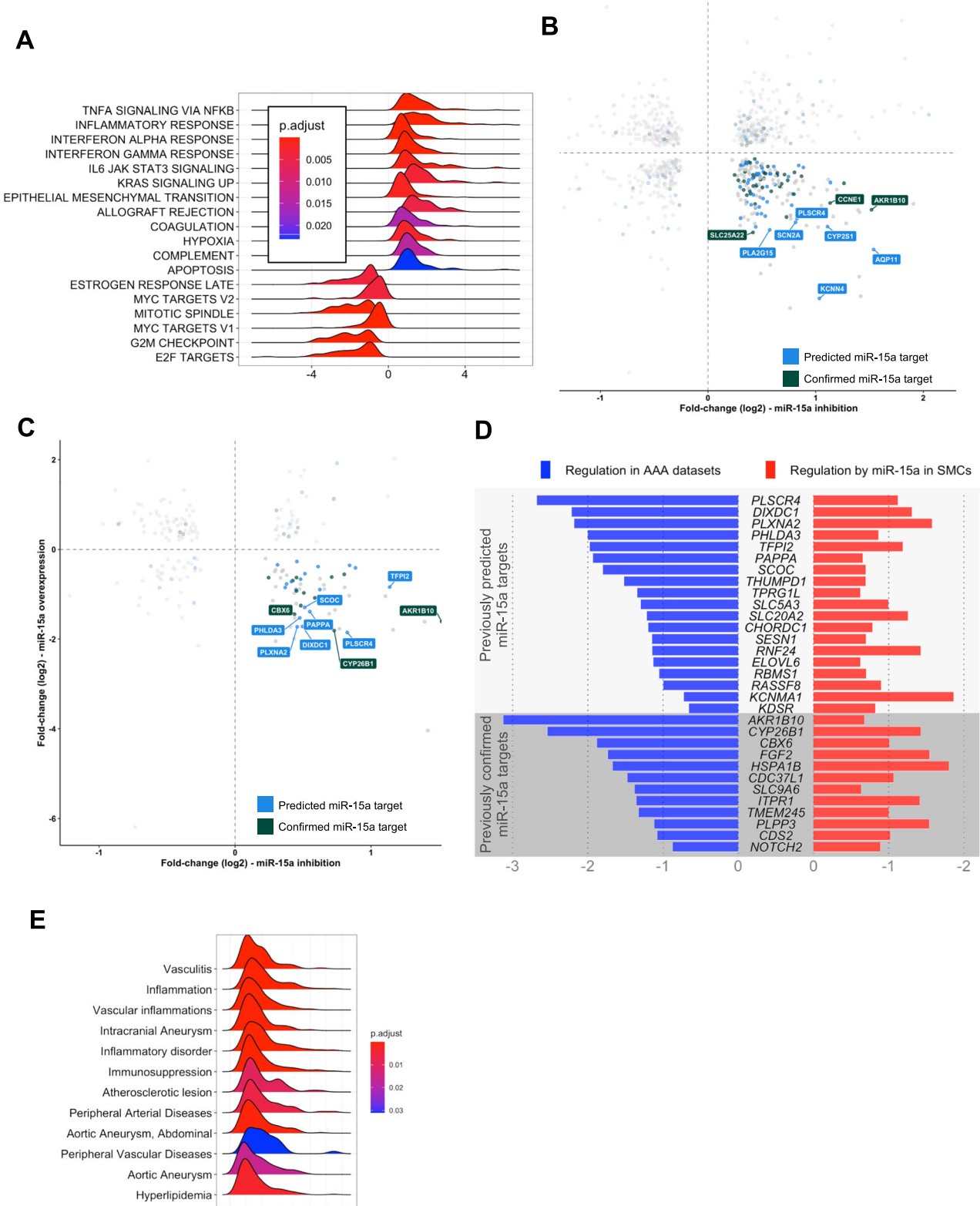

**Fig. 5 | Effects of miR-15a modulation in hAoSMCs on gene-sets of relevance to human AAA pathology. A** Ridge plot of GSEA analysis results using MSigDB-H (hallmark) gene-sets. **B** Fold change plot showing genes significantly down-regulated (FDR < 0.05, fold change (log2) < 0) in hAoSMCs treated with miR-15a-mimic and significantly up-regulated (FDR < 0.05, fold change (log2) > 0) in hAoSMCs treated with miR-15a-inhibitor. Highlighted in blue are predicted miR-15a targets (present in DIANA or miRDB), in green experimentally confirmed miR-15a targets (present in miRTarBase). Labeled are top 10 differentially regulated genes with largest difference in expression between miR-15a-inhibitor and mimic treatments. **C** Data as in (**B**), only genes also down-regulated in human AAA datasets are shown. **D** Bar plot showing regulation of genes in **C** in AAA datasets (left side) and miR-15a-modulated hAoSMCs (right side; fold change between miR-15a-inhibitor and mimic treatments). Genes labels are colored in same colors as highlighted genes in (**B**). **E** Ridge plot of GSEA analysis results in manually selected DisGeNET gene-sets of relevant diseases.

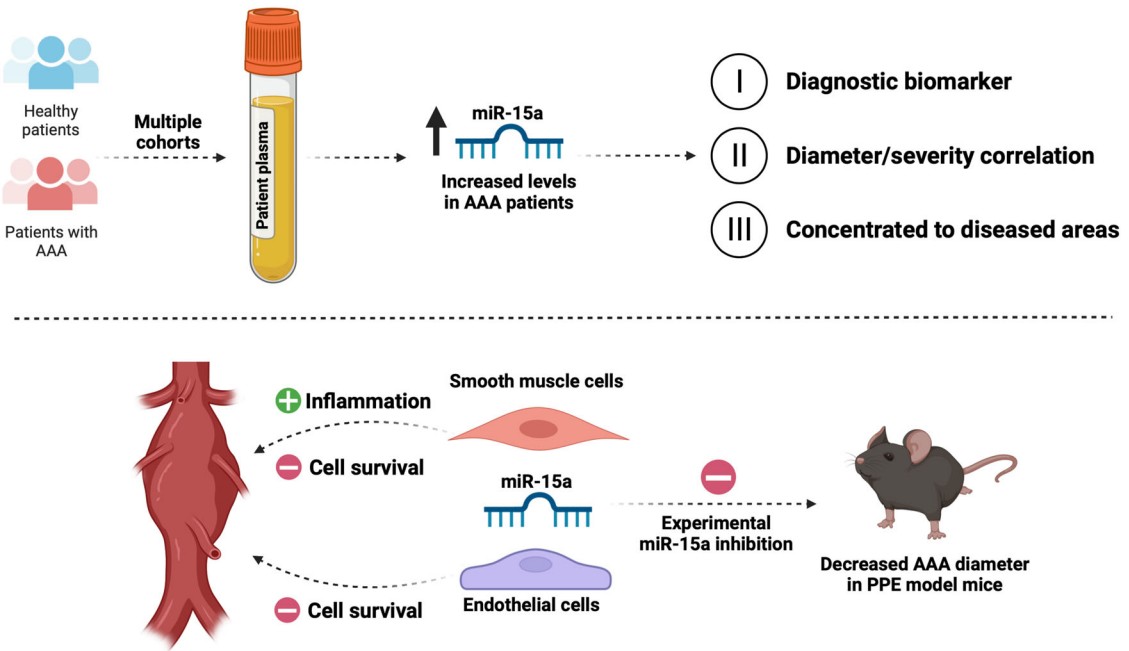

**Fig. 6 | miR-15a is involved in regulation of known mechanisms in AAA pathogenesis and its circulating levels are dysregulated among AAA patients.** Biofluids from multiple cohorts of AAA patients were analyzed, discovering differences in expression of miR-15a and thus revealing a biomarker with potential diagnostic implications, but also shown to be concentrated in diseased areas. miR-15a has been described to play a role in the fate of aortic smooth muscle and endothelial cells. Our studies in a mouse AAA model show that its inhibition can limit growth of experimentally induced aortic aneurysms. Created in BioRender (https://BioRender.com/pg73t6h).

15a levels and larger AAA diameters. In one of these cohorts collected material is serum, rather than plasma, allowing us to verify the performance of miR-15a in a second blood-derived biofluid. The choice between plasma and serum for miRNA biomarker studies has been extensively discussed before. In brief, serum samples have been reported to contain slightly higher levels of miRNAs, although having undergone coagulation they are more sensitive to perturbations stemming from platelet miRNA content[72]. Further, we confirm miR-15a to be expressed in VSMCs of tunica media, where it has a negative effect on their viability and survival, and additionally we show that inhibition of miR-15a inhibits in vivo murine AAA growth. In addition, KLF4, a transcription factor involved in AAA pathology and involved in phenotype regulation of VSMCs[73], has been reported to directly regulate transcription of miR-15a, through which it exerts its anti-proliferative and anti-angiogenic properties on ECs and VSMCs[51]. Together, these findings present a strong argument for the logic that increased, rather than decreased, circulating miR-15a levels are connected to and likely involved in AAA pathology.

*SMPD1*, a gene which codes for acid sphingomyelinase (ASM), was one of the most down-regulated predicted target genes upon miR-15a overexpression in hAoSMCs (Supplementary Fig. 1A). ASM, a mediator of ceramide metabolism[74], is a known crucial regulator of atherogenesis[75]. It is required for autophagy maturation of human VSMCs and controls their differentiation into a more contractile phenotype, leading to a net anti-atherogenic effect[76,77] and its deficiency has been linked to increased incidence of human coronary atherosclerosis[78,79]. *Smpd1*$^{-/-}$ VSMCs, upon stimulation with PDGF-BB, undergo morphological changes and differentiate into a pro-fibrotic and pro-inflammatory myofibroblast-like phenotype, with enhanced TGF-β1 secretion, collagen deposition, increased adhesion to monocytes, and massive production of IL-6 and IL-18[80]. This contrasts with *Smpd1*$^{+/+}$ VSMCs which instead acquire a more conventional synthetic phenotype[80]. This effect seems to be at least partially mediated by absent inhibition of PI3K/AKT-signaling by ceramide-activated PP2A[80]. Activation of PI3K/AKT-signaling is known to be increased in AAA-tissues[81]. Although our miR-15a overexpressing hAoSMCs were not PDGF-

BB treated, we still observed an overall pro-inflammatory, anti-proliferative and pro-apoptotic response, in line with the above-mentioned findings. Another study, looking at VSMC overexpression of *Smpd1* in mice, observed increased medial calcification and stiffness of the arterial wall[82]. Vascular wall calcification has been described a potential risk factor for aneurysm rupture[83], meaning SMPD1 might not necessarily only have beneficial roles from a AAA disease perspective.

Further, decreased levels of circulating miR-15a have been associated with type 2 diabetes and negatively correlated with fasting glucose levels[84]. Its expression was found to be increased in high-glucose conditions which via inhibition of UCP2 leads to increased insulin biosynthesis[85]. Diabetes is known to have a protective effect on AAA and its progression[86]. Intriguingly, diabetic patients have increased expression of ceramides in several tissues and in plasma[87], where ceramides are suspected to play a role in development of insulin resistance[88], also in part mediated via the ceramide/PP2A-AKT axis[89]. Consequently, the markedly different roles of miR-15a and SMPD1 in diabetes and AAA, as well as changes in levels of circulating miR-15a, could explain parts of the interplay between the two diseases. It needs to be noted that none of the studied patient cohorts were matched for prevalence of type 2 diabetes (Table 1), and thus might be a potential source of confounding. However, we did not observe any differences in plasma miR-15a levels between AAA patients with and without diabetes (Supplementary Figs. 2D and 5C, D), or between other patient characteristics (Supplementary Figs. 4 and 5).

MMPs are key players in AAA pathology and the inhibition of MMPs is a successful strategy for limiting experimental AAA progression[90]. *RECK* is a key inhibitor of MMPs known to be regulated by miR-205/712 in AAA[24]. Its down-regulation due to direct inhibition by miR-15a leads to increased secretion of MMP9 in neuroblastoma cells[91]. In line with this, we noted *RECK* as down-regulated in hAoSMCs upon miR-15a overexpression. In addition, we found *TFPI2*, another inhibitor of MMPs and predicted target of miR-15a, to be increased upon miR-15a inhibition in hAoSMCs. Diminished levels of TFPI2 in atheromatous plaques are suspected to contribute to their vulnerability[92].

Additional factors driving AAA progression are decreased survival of VSMCs and dedifferentiation of VSMCs into non-contractile phenotypes[93]. miR-15a has been described to be up-regulated in VSMCs extracted from AAA patients and regulate their apoptosis through targeting of *CDKN2B*[52]. In our hands *CDKN2B* was indeed down-regulated upon miR-15a over-expression in hAoSMCs (Supplemental Data 2), and this condition had a negative effect on their growth. Some of the most regulated genes in miR-15a-modulated hAoSMCs were *FGF2 (bFGF)* and *NOTCH2*, respectively down/up-regulated upon miR-15a overexpression/inhibition. FGF2 is known as beneficial in AAA through its positive effects on SMC proliferation[94], and *NOTCH2* is required for contractile differentiation of VSMCs[95].

Among other differentially regulated genes identified in miR-15a-modulated hAoSMCs, several are involved in regulation of VEGF-signaling —RASSF8[96], STC1[97] and CDS-2[98]. VEGF-signaling and resulting angiogenesis regulation are thought to play major roles in AAA progression, and our group, as well as other groups have recently reported on beneficial effects of VEGF/VEGF-receptor inhibition in experimental AAA[99,100]. Interestingly, miR-15a is itself a direct inhibitor of *VEGF* and has been described to inhibit peritoneal fibrosis in peritoneal dialysis patients[101]. Similar anti-fibrotic effects in peritoneal fibrosis have been attributed to miR-29b[102], a miRNA which is known to be detrimental to AAA and proposed as a potential strategy of limiting AAA growth[14,15,103]. In retinal ECs, miR-15a has been shown to inhibit angiogenesis and decrease permeability of the intimal layer via its regulation of *VEGF* and *TGF-β3*[104]. We found *VEGF (VEGF-A)* to be increased in hAoSMCs upon inhibition of miR-15a, but upon over-expression its levels were unchanged (Supplemental Data 2).

The different aspects of miR-15a effect on VEGF-signaling underlines the notion that the gene expression changes caused by its modulation should be viewed in the broader perspective of disease phenotype regulation, rather than on/off regulation of specific genes or upstream regulators of signaling-networks. Most of the above-described effects of miR-15a on key AAA genes are likely detrimental to AAA, however we believe the overall effect of miR-15a on AAA phenotype is unlikely to be solely through these effects. While there it is often a desire to find specific mechanisms of action, it must be noted that miRNAs in general tend to have many gene targets. Specific mechanisms can of course be mediated through specific key genes, but they need to be viewed in a perspective of the complex biological networks that miRNAs have the ability to regulate, where effects of one gene may be counterbalanced by changes in others. We therefore chose to focus more on the overall effect that miR-15a might have on AAA, through integration of target prediction databases and previously published datasets of differentially regulated genes in AAA disease. Through GSEA analyses on RNA-seq data from experimentally modulated hAoSMCs, we show that its increased expression of miR-15a leads to net down-regulation genes involved in cell cycle progression and to net up-regulation of genes involved in stress response, thus inducing a more inflammatory phenotype. Progression of AAA is characterized by inflammation and ECM degeneration[105]. Trans-differentiation of medial VSMC into synthetic fibroblast-like or inflammatory macrophage-like phenotypes, as well as their decreased survival are known to play major roles in AAA development and progression[106].

To conclude, in the first sufficiently powered study investigating miRNAs as plasma biomarkers for AAA, we found 12 circulating miRNAs to be significantly dysregulated in AAA. Among these, miR-15a was confirmed to be dysregulated in several other independent cohorts confirmed to be associated with underlying AAA pathologies correlating to AAA diameters in those patients. Experimental inhibition of miR-15a was shown to attenuate AAA growth in an experimental murine model (PPE). RNA sequencing of miR-15a-modulated hAoSMCs revealed a potential large target network, and miR-15a overexpression had net pro-inflammatory, pro-apoptotic and anti-proliferative effects. Due to its direct involvement in AAA pathogenesis, miR-15a is potentially one of the most promising AAA biomarkers to date. Thanks to above-mentioned properties, miR-15a does

seem to be a potential risk-stratifying biomarker of a disease in which such biomarkers have thus far been largely lacking. To further evaluate this potential, novel prospective clinical studies will be required. We believe that any such upcoming studies, looking to evaluate potential biomarker combinations to better diagnose and risk-stratify AAA disease, should include measurements of miR-15a in their study design.

## Data availability
Key data from the analyses in this article is available in the online Supplementary Material and all remaining data will be shared on reasonable request to the corresponding author. RNAseq data generated as part of this manuscript has been deposited in the NCBI Gene Expression Omnibus (GSE266337). RNAseq data together with Supplementary Data 2 are the source data for Figs. 4 and 5. Supplementary Data 1 is the source data for Fig. 1A–C.

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

## Acknowledgements

This work was supported by the Swedish Heart-Lung-Foundation (20180680), the Swedish Research Council (Vetenkapsrådet, 2019-01577), the European Research Council (ERC-StG NORVAS), a DZHK Junior Research Group (JRG_LM_MRI), the SFB1123 and TRR267 of the German Research Council (DFG), the National Institutes of Health (NIH; 1R011HL150359-01), the Bavarian State Ministry of Health and Care through the research project DigiMed Bayern—all to L.M. G.W. has received support from The Committee for Doctoral Education at Karolinska Institutet (CSTP/Research Internship programmes).

## Author contributions

L.M., G.W. and S.E. conceived the study, with critical input from J.R., C.B., G.J.B., J.M.S., P.T. and F.W.A. G.W., E.C., A.Bu., S.E., H.J., A.Bä. and V.P. designed and conducted the majority of experiments. G.W., A.Bu., H.J. and A.Bä. designed and conducted animal experiments. T.K. and N.S. assisted with experiments and data collection. G.W. conducted bioinformatic analyses. M.L.L. and J.R. provided data and contributed to experimental design and data interpretation. R.H., C.B., G.J.B., J.P.G.S., H.H.E., R.A.B. and F.W.A. provided access to patient materials. G.W. wrote the manuscript, with critical input from L.M., J.S., E.C., S.E., A.Bu., C.B. and J.P.G.S. All authors have read and approved the final manuscript.

## Funding

## Competing interests

L.M. is a scientific advisor for Novo Nordisk (Malov, Denmark), Angiolutions (Hannover, Germany), and received research funds from Novo Nordisk (Malov, Denmark) and Roche Diagnostics (Rotkreuz, Switzerland). All remaining authors (G.W., E.C., A.Bu., S.M.E., H.J., M.L.L., T.K., A.Bä., V.P., J.R., R.H., C.B., G.J.B., J.P.G.S., N.S., P.S.T., F.W.A.) declare no competing interests.

## Additional information

[1]Department of Medicine, Solna, Karolinska Institutet, Stockholm, Sweden. [2]Function Perioperative Medicine and Intensive Care, Karolinska University Hospital, Stockholm, Sweden. [3]Division of Vascular and Endovascular Surgery, Department for Visceral-, Thoracic and Vascular Surgery, Medical Faculty Carl Gustav Carus and University Hospital, Technische Universität Dresden, Dresden, Germany. [4]Department of Molecular Medicine and Surgery, Karolinska Institutet, Stockholm, Sweden. [5]Veterans Affairs Palo Alto Health Care System, Palo Alto, CA, USA. [6]Stanford Cardiovascular Institute, Stanford University, Stanford, CA, USA. [7]Department for Vascular and Endovascular Surgery, Klinikum Rechts der Isar, Technical University of Munich, Munich, Germany. [8]German Center for Cardiovascular Research (DZHK), Partner Site Munich Heart Alliance, Berlin, Germany. [9]Division of Vascular Surgery, Department of General Surgery, Medical University of Vienna, Vienna, Austria. [10]Department of Vascular Surgery, University Medical Center Utrecht, Utrecht, The Netherlands. [11]Department of Cardiology, Experimental Cardiology Laboratory, University Medical Center Utrecht, Utrecht University, Utrecht, The Netherlands. [12]Department of Physiology, VU University Medical Center in Amsterdam, Amsterdam, The Netherlands. [13]German Center for Cardiovascular Research (DZHK), Partner Site Rhein/Main, Berlin, Germany. [14]Institute of Cardiovascular Regeneration, Goethe University, Frankfurt am Main, Germany. [15]Department of Cardiology, Amsterdam Cardiovascular Sciences, Amsterdam University Medical Centre, University of Amsterdam, Amsterdam, The Netherlands. [16]Institute of Health Informatics, University College London, London, UK. [17]The National Institute for Health Research University College London Hospitals Biomedical Research Centre, University College London, London, UK. ✉e-mail: lars.maegdefessel@tum.de

