## [Transparent Peer Review file · Communications Medicine]

microRNA-15a-5p mediates abdominal aortic aneurysm progression and serves as a potential diagnostic and prognostic circulating biomarker

Corresponding Author: Professor Lars Maegdefessel

Version 0:

Reviewer comments:

Reviewer #1

(Remarks to the Author)

This is a wonderful use of translational model with support from relevant clinical data from robust resources (both clinical studies and data bases).

The work is well performed with appropriate controls and includes in vivo and in vitro testing.

The manuscript can be improved by:

- 1) clarifying the male/female composition of the clinical study data
- 2) discussing/clarifying how these microRna pathways are affected by sex/gender (particularly in the peri- and post-menopausal time frame)
- 3) adding as a limitation the focus on male sex. I understand that there is a male predominance of this disease, but with aging/survival, women screening is equally important.

Reviewer #2

(Remarks to the Author)

The authors examined the diagnostic and prognostic value of circulating microRNAs in abdominal aortic aneurysm (AAA) disease. The results suggest that miR-15a is a potential biomarker for AAA. Through in vivo studies and based on its target profile, we showed that miR-15a is involved in the pathogenesis of AAA and may be useful not only for treatment but also for risk stratification of AAA patients and identification of candidates for early AAA repair. The experimental methods are accurate and the conclusions are logical. Only the following point needs clarification.

1. miR-inhibitor is not easily accessible to smooth muscle cells. please show that anti-miR-15 is introduced into living smooth muscle cells in the experiment in Figure 3.

Version 1:

Reviewer comments:

Reviewer #1

(Remarks to the Author)

responsive to critiques
manuscript is improved with no further recommendations

Reviewer #2

(Remarks to the Author)

The manuscript has been improved. No further comments.

We thank the reviewers for their comments and their positive response to our manuscript. Below, the specific concerns are addressed on a point-by-point basis. We have also adjusted our manuscript text to better reflect these concerns.

Reviewer #1 (Remarks to the Author):

This is a wonderful use of translational model with support from relevant clinical data from robust resources (both clinical studies and data bases).

The work is well performed with appropriate controls and includes in vivo and in vitro testing.

The manuscript can be improved by:

1) clarifying the male/female composition of the clinical study data

We thank the reviewer for giving us the opportunity to comment on this: AAA is 4-6 times more common in men than in women, men have a 5-10 year earlier onset of disease and a continuously higher prevalence in all age groups. Population based screening is ongoing for 65-year old men in Sweden. This enhances the possibility to include and analyze more patients with small aneurysms, however these will be men. Women are in minority both in the outpatient service but also in the treatment groups, since they are fewer and 5-10 years older. These impressive sex-differences do influence the biobanks collected worldwide in aneurysm cohorts. The included biobank material reflects these sex differences.

The male/female composition of the studied patient cohorts is provided in Table 1. To further clarify these challenges and limitation we also chose to include the above statement in the Study Population subsection of the manuscript.

2) discussing/clarifying how these microRna pathways are affected by sex/gender (particularly in the peri- and post-menopausal time frame)

Women develop AAA in their post-menopausal period. Mean age for menopause is 49-51 years depending on smoking habits. The development of AAA starts after 60 years of age. It is relatively unknown if the premenopausal sex hormone levels would influence AAA development. (Hultgren *et al.* Ann Cardiothorac Surg 2023, doi: 10.21037/acs-2023-adw-17). At this point we can only speculate that microRNA pathways in elderly women and men with AAA show less differences than AAA patients vs. controls. But obviously further studies with a specific focus on this would need to be conducted.

3) adding as a limitation the focus on male sex. I understand that there is a male predominance of this disease, but with aging/survival, women screening is equally important.

We agree with the reviewer on this. The vast sex differences in the AAA-patient groups differ from all other cardiovascular patient groups. The male predominance with early onset, 5-fold higher incidence rate and need for earlier surgery than in women does largely explain

why the biobanks around the world consist predominantly of material collected in men. There is indeed a need to continuously explore the impressive sex differences in the epidemiological characteristics of AAA, and by improving our biological understanding of disease development this can be obtained. Our collaborating centers (manly Rebecka Hultgren at the Karolinska Institutet and University Hospital in Stockholm) are presently working on increasing the inclusion of women in biobanks by developing targeted screening programs, which ideally will contribute to better understanding this aspect of disease development, progression and provide better treatment guidance.

Reviewer #2 (Remarks to the Author):

The authors examined the diagnostic and prognostic value of circulating microRNAs in abdominal aortic aneurysm (AAA) disease. The results suggest that miR-15a is a potential biomarker for AAA. Through *in vivo* studies and based on its target profile, we showed that miR-15a is involved in the pathogenesis of AAA and may be useful not only for treatment but also for risk stratification of AAA patients and identification of candidates for early AAA repair. The experimental methods are accurate and the conclusions are logical. Only the following point needs clarification.

1. miR-inhibitor is not easily accessible to smooth muscle cells. please show that anti-miR-15 is introduced into living smooth muscle cells in the experiment in Figure 3.

While this validation was not performed for the present study, our group (among others) has vast experience of antisense oligonucleotides into SMCs *in vivo*. In fact, this was recently validated in a AAA-manuscript by Winter *et al.* (Suppl. Fig 1F; *Mol Ther* 2023, doi:10.1016/j.ymthe.2023.04.020).

For the editor's and reviewers' convenience, a previously unpublished image of scrambled antisense oligonucleotide delivery is also presented as Reviewer Figure 1 below. Here, uptake into the medial layer of a growing aneurysm is presented.

Reviewer Figure 1.